# TIM3 Expression in Anaplastic-Thyroid-Cancer-Infiltrating Macrophages: An Emerging Immunotherapeutic Target

**DOI:** 10.3390/biology11111609

**Published:** 2022-11-03

**Authors:** Luz Maria Palacios, Victoria Peyret, María Estefania Viano, Romina Celeste Geysels, Yair Aron Chocobar, Ximena Volpini, Claudia Gabriela Pellizas, Juan Pablo Nicola, Claudia Cristina Motran, María Cecilia Rodriguez-Galan, Laura Fozzatti

**Affiliations:** 1Departamento de Bioquímica Clínica, Facultad de Ciencias Químicas, Universidad Nacional de Córdoba, Córdoba 5000, Argentina; 2Centro de Investigaciones en Bioquímica Clínica e Inmunología, Consejo Nacional de Investigaciones Científicas y Técnicas (CIBICI-CONICET), Córdoba 5000, Argentina; 3Hospital de Endocrinología y Metabolismo Dr. Arturo Oñativia, Salta 4400, Argentina

**Keywords:** anaplastic thyroid cancer, tumor-associated macrophages, M2-like macrophage polarization, TIM3, immunotherapy

## Abstract

**Simple Summary:**

Anaplastic thyroid cancer (ATC) is a highly lethal type of cancer. Patients rarely survive beyond 6 months after diagnosis because of its aggressiveness and lack of an effective treatment. Therefore, there is an urgent need to identify new biological targets that can be translated into novel clinical approaches. Regarding this, ATC is heavily infiltrated with tumor-associated macrophages (TAMs), making them attractive therapeutic targets. Immunotherapy is being explored for patients with ATC, although its efficacy remains still limited. Hence, we consider that targeting immune checkpoints in TAMs could be a promising approach for ATC. We have recently shown that treatment of THP-1 cells (human monocyte-like cell line) with conditioned media derived from ATC cells induced their activation toward a pro-tumoral phenotype along with upregulation of the immune checkpoint marker TIM3. Experiments using TIM3-blocking antibodies partially reversed these effects, suggesting the role of this receptor in the activation and pro-tumoral effects of TAMs in ATC in vitro. Here, we further validated those observations during the development of ATC in vivo. We have detected, for the first time, the presence of the immune checkpoint in TAMs in ATC xenograft tumors. Therefore, TIM3 should be considered as an immunotherapeutic target for ATC.

**Abstract:**

Anaplastic thyroid cancer (ATC) is a clinically aggressive form of undifferentiated thyroid cancer with limited treatment options. Immunotherapy for patients with ATC remains challenging. Tumor-associated macrophages (TAMs) constitute over 50% of ATC-infiltrating cells, and their presence is associated with a poor prognosis. Consequently, the development of new therapies targeting immune checkpoints in TAMs is considered a promising therapeutic approach for ATC. We have previously shown that soluble factors secreted by ATC cells induced pro-tumor M2-like polarization of human monocytes by upregulating the levels of the inhibitory receptor TIM3. Here, we extended our observations on ATC-cell-induced xenograft tumors. We observed a large number of immune cells infiltrating the ATC xenograft tumors. Significantly, 24–28% of CD45^+^ immune cells were macrophages (CD11b^+^ F4/80^+^). We further showed that 40% of macrophages were polarized toward a M2-like phenotype, as assessed by CD206 expression and by a significant increase in the Arg1/iNOS (M2/M1) ratio. Additionally, we found that ATC xenograft tumors had levels of TIM3 expression when determined by RT-PCR and immunofluorescence assays. Interestingly, we detected the expression of TIM3 in macrophages in ATC tumors by flow cytometry assays. Furthermore, TIM3 expression correlated with macrophage marker expression in human ATC. Our studies show that TIM3 is a newly identified immune checkpoint in macrophages. Since TIM3 is known as a negative immune regulator, it should be considered as a promising immunotherapeutic target for ATC.

## 1. Introduction

Thyroid cancer (TC) is the most frequently occurring endocrine cancer [1]. Among the different types of TC, anaplastic thyroid cancer (ATC) is a rare malignancy but remains one of the most lethal types of human cancer with a median patient survival of ≈5 months after diagnosis [2,3,4]. Currently, effective treatment options for ATC patients are limited [3,4]. Therefore, finding new therapeutic targets for these tumors that can be translated into novel clinical approaches is urgently needed.

Recent evidence suggests that contributions from the tumor microenvironment (TME) influence cancer cell behavior, disease progression, and response to therapy in many different types of neoplasia [5,6]. The TME is an active and dynamic component supporting tumor growth, contributing to tumor cell survival that comprises a variety of non-tumor cells, including fibroblasts, immune cells, and endothelial cells, among others [5,6]. However, the contribution of the TME to TC progression remains largely unexplored. We previously described the dynamic interaction that exists between ATC cells and human thyroid fibroblasts that potentiates TC progression [7,8].

Macrophages are one of the major components of the TME [9,10]. In the TME, macrophages are called tumor-associated macrophages (TAMs) and are derived mainly from circulating monocytes [11]. Local signals derived from tumor cells enforce distinct functional activation states of macrophages, placing them along a spectrum of functional states, with the pro-inflammatory/anti-tumor (M1) macrophages at one extreme and the anti-inflammatory/pro-tumor (M2) macrophages at the opposite side [12]. In advanced solid cancers, the secretion of cytokines and tumor signals are commonly thought to polarize macrophages toward the M2-like phenotype linked with tumor progression and suppression of tumor-specific immunity [11]. Importantly, TAMs represent more than 50% of the cells in the ATC [13,14,15,16], which makes them attractive therapeutic targets [17]. However, the way in which TAMs acquire this M2-like phenotype within the TME and influence tumor progression in ATC remains largely undefined.

We recently provided valuable insights into the processes in which soluble factors produced by ATC cells induce pro-tumor M2-like polarization of human monocytes through T-cell immunoglobulin and mucin-domain containing protein-3 (TIM3) [18]. In the present study, we demonstrated a large recruitment of M2-like macrophages into ATC-cell-derived xenograft tumors, recapitulating ATC in humans. We further showed, for the first time, the expression of TIM3 on TAMs in ATC-cell-induced xenograft tumors. In addition, and in public datasets, we found that TIM3 expression positively correlated with the expression of macrophage markers in human ATC.

Despite the abundant immune cell infiltrate observed in ATC, less than 20% of the patients benefit from therapy with anti-PD-1 antibodies [19], highlighting the importance of defining new immune checkpoint inhibitors as well as targeting innate immune cells to improve the anti-tumor responses in patients with ATC. Therefore, our findings revealing the expression of TIM3 in ATC-infiltrating macrophages may contribute to the development of novel immunotherapy strategies for ATC.

## 2. Materials and Methods

### 2.1. Mice and Cell Lines

Female non-obese diabetic severe combined immunodeficient (NOD/SCID) mice were maintained under specific pathogen-free conditions. Animal care was provided in accordance with the procedures outlined in the Guide for the Care and Use of Laboratory Animals (National Institutes of Health, NIH). The experimental protocols were approved by the Institutional Animal Care and Use Committee. Our animal facility obtained NIH animal welfare assurance (assurance no. A5802-01, Office of Laboratory Animal Welfare; NIH, Bethesda, MD, USA).

Human ATC cell line 8505C was obtained from the European Collection of Authenticated Cell Cultures and cultured in Dulbecco’s modified Eagle’s medium (DMEM) (Gibco, Thermo Fisher Scientific, Waltham, MA, USA), as previously reported [7]. The ATC-derived C643 cell line was obtained from the University of Colorado Cancer Center Cell Bank and maintained in RPMI 1640 medium (Gibco, Thermo Fisher Scientific, Waltham, MA, USA). RAW 264.7 was obtained from the American Type Culture Collection (ATCC) and was cultured in DMEM (Gibco, Thermo Fisher Scientific, Waltham, MA, USA). Human ATC cells were authenticated by short tandem repeat (STR) profiling analysis [7,18]. All cell lines were cultured in medium supplemented with 10% fetal bovine serum (FBS, Hyclone, Logan, UT, USA), penicillin/streptomycin, and L-glutamine (Gibco, Thermo Fisher Scientific, Waltham, MA, USA).

### *2.2. In Vivo* Mouse Xenograft Model

We subcutaneously inoculated 10–12-week-old NOD/SCID mice into the lower right flank with 5 × 10^6^ 8505C or C643 cells resuspended in 100 µL of phosphate-buffered saline (PBS), as described previously [20]. The tumor size was monitored with a caliper every 1–4 days. Mice were ethically sacrificed under isoflurane anesthesia followed by cervical dislocation when the median tumor size reached approximately 800–1300 mm^3^. Tumor weight was determined after the dissection of xenografted tumor tissues from mice at the study endpoint. Dissected tumors were collected for further analysis.

### 2.3. Flow Cytometry Analysis

Tumors were mechanically disrupted and resuspended at 1 g of tumor per 7 mL of PBS. Samples were first washed with PBS and then stained with a Zombie Aqua™ Fixable Viability Kit (Biolegend, San Diego, CA, USA) for 15 min at room temperature for the exclusion of dead cells. Expression of surface markers was assessed by staining with appropriate combinations of the following fluorescent-labeled monoclonal antibodies for 30 min at 4 °C: CD45 (#103116) from Biolegend and CD11b (#67-0112-80), F4/80 (**#**25-4801-82), CD206 (#46-2061-82), and TIM3 (**#**12-5870-82), which were obtained from eBioscience (San Diego, CA, USA). Following two washings, cells were resuspended in FACS buffer and analyzed in the flow cytometer. To detect intracellular Arginase 1 (Arg1) and inducible nitric oxide synthase (iNOS), cells were stained for surface markers, washed, and fixed with IC Fixation Buffer (eBioscience, San Diego, CA, USA) for 90 min at 4 °C. Cells were washed with Permeabilization Buffer (eBioscience) and incubated for 30 min with the same buffer. Cells were centrifuged and incubated with the Arg1 eF450 (#48-3697-80) and iNOS PE anti-mouse (#12-5920-80) antibodies (eBioscience, San Diego, CA, USA) for 45 min at 4 °C, washed, and resuspended in FACS buffer for flow cytometry analysis in a BD FACS Canto II or BD LSR Fortessa X-20 flow cytometers (BD Biosciences San Jose, CA, USA). Data analysis was performed by FlowJo software (BD Life Sciences, Ashland, OR, USA). The gating strategies for the analysis of cell populations are shown in Figure 1.

### 2.4. RNA Isolation and Reverse Transcription (RT)-PCR

Total RNA was extracted from frozen sections of xenograft tumors using TRIzol (Sigma-Aldrich, St. Louis, MO, USA) according to the manufacturer’s instructions. cDNA synthesis and PCR were performed as previously described [7,18,21,22]. Gene specific primer sets were as follows: mouse TIM3 (271 bp), 5′-ACTGGTGACCCTCCATAATAACA-3′ (forward) and 5′-ATTTTCCTCAGAGCGAATCCT-3′ (reverse); mouse β-actin (223 bp), 5′-CTACAATGAGCTGCGTGTGG-3′ (forward) and 5′-GGGCACAGTGTGGGTGAC-3′ (reverse). RT-PCR products were resolved by electrophoresis on 2% agarose gels containing ethidium bromide.

### 2.5. Immunofluorescence Staining

Tumors were harvested, fixed with 4% paraformaldehyde, embedded in CryoplastR (Biopack), and cut into 5 µm thick sections using a Cryostat 0620E (Thermo Scientific Shandon, USA). Sections were rehydrated in Tris buffer 50 mM, heated to 90 °C in antigen retrieval solution (citrate buffer, pH 6) for 20 min, and then blocked for 90 min with 10% rat serum-Tris buffer. After blocking, slides were incubated overnight at 4 °C with PE-labeled anti-mouse F4/80 (1:250 dilution, #123110 BioLegend, clone BM8) or PE-labeled anti-mouse TIM3 (1:50 dilution, #119704, BioLegend, clone RMT3-23) in blocking buffer. Sections were counterstained with 4,6-diamidino-2-phenylindole (DAPI). Slices were mounted with FluorSave (Merck Millipore, Burlington, MA, USA). Images were collected with a Leica microscope (DMi8).

### 2.6. Gene Expression Analysis from Public Datasets

Affymetrix-GPL570-platform-based microarray datasets containing ATCs (GSE29265, GSE33630, GSE76039, and GSE65144) and normal thyroid tissue (GSE3467, GSE3678, GSE6004, GSE29265, GSE33630, GSE53157, GSE35570, and GSE60542) were downloaded from the NCBI Gene Expression Omnibus database (http://www.nibi.nih.gov/geo/ accessed on 1 March 2022). Data were normalized using the GeneChip Robust Multiarray Averaging method, and a matrix of 171 normal tissues and 52 ATCs was generated. Genes were annotated using hgu133plus2.db package version 3.2.3 [18].

### 2.7. Statistical Analysis

Statistical analysis was conducted using GraphPad Prism 6.01 software (GraphPad Software, San Diego, CA, USA). Comparisons between two groups were carried out using unpaired Student’s t-test or non-parametric Mann–Whitney U-test. Differences were considered significant at *p* < 0.05.

## 3. Results

### 3.1. Immune Cells Were Recruited to *ATC Xenograft* Tumors

Solid tumors are commonly infiltrated by immune cells, which can play a crucial role in dictating tumor progression. Despite the importance of the immune microenvironment, few studies have explored immune cell infiltration in TC. To investigate the cellular components of the immune system within the thyroid TME of ATC, two mouse xenograft models were generated by subcutaneous inoculation of 8505C and C643 ATC cells into NOD/SCID mice. The means of the tumor growth curves for each xenograft mouse are shown in Figure 2A. We observed that tumors derived from C643 cells grew significantly faster than tumors derived from 8505C cells. Due to the difference in tumor growth between both groups, and in order to obtain comparable tumor sizes at the end of the experiments, 8505C cells were inoculated into mice 2 weeks before C643 cells. All mice were euthanized at the same time at the end of the experiments. Therefore, we found that the mean weight of C643 xenograft tumors showed a trend toward increase compared to 8505C xenograft tumors; however, they did not reach statistical significance at the study endpoint (Figure 2B). We quantified the overall number of immune cells within the TME on whole *xenograft* tumors by flow cytometry analysis. A large quantity of immune infiltrate was observed in both xenograft models. Conversely, we found 11% of CD45^+^ cells (leukocytes) within 8505C xenograft tumors (Figure 2C), and C643 xenograft tumors were largely infiltrated, showing 58% CD45^+^ cells (Figure 2D). Interestingly, while we observed a great proportion of dead CD45^neg^ cells in both tumor types, most CD45^+^ cells were alive (Figure 2C,D). Quantification of total CD45^+^ living cells is shown in Figure 2E.

A hallmark of ATCs is their heavy infiltration with M2-like TAMs, representing more than 50% of the tumor volume [13,14,15,16]. The presence of TAMs is associated with a worse prognosis in ATC [13]. Using flow cytometry analysis with the macrophage lineage markers F4/80 and CD11b, we first determined the percentage of infiltrating macrophages into the ATC xenograft tumors (Figure 3A–C). Even though C643 xenograft tumors presented higher number of CD45^+^ cells than 8505C xenograft tumors, the frequencies of double-positive CD11b^+^ F4/80^+^ cells were similar between both tumor types (24% were found positive for CD11b and F4/80 in 8505C xenograft tumors (Figure 3A), while 27% were detected positive for CD11b and F4/80 in C643 xenograft tumors (Figure 3B)). Quantification of the percentage of CD11b^+^ and F4/80^+^ cells is shown in Figure 3C. The presence of macrophages was further validated in 8505C xenograft tumors by using immunofluorescence analysis of tumor sections with the F4/80 antibody (Appendix A), which confirmed macrophage infiltration into the tumors. Similar results were observed in C643 xenograft tumors (Appendix A).

We have recently reported that ATC-cell-derived conditioned media (CM) were able to induce tumor-promoting M2-like polarization of human monocytes [18]. Therefore, in order to validate our in vitro studies, we next determined whether the macrophages in ATC xenograft tumors were similarly polarized toward the M2-like phenotype. We first evaluated the expression levels of the mannose receptor CD206, a marker expressed by M2 macrophages [23]. We found that in 8505C xenograft tumors, 42% of the macrophages expressed CD206 (CD11b^+^ F4/80^+^ CD206^+^ cells, Figure 3D), while 40% of them expressed CD206 in C643 xenograft tumors (CD11b^+^ F4/80^+^ CD206^+^ cells, Figure 3E), demonstrating a similar frequency of M2-like macrophages in both tumor types (Figure 3F). Interestingly, these results are in line with our previous observations showing an increased expression of CD206 following the activation of human monocytes with ATC cell-derived CM [18]. It has been described that M2 macrophages also display enhanced expression of arginase 1 (Arg1) with decreased expression of inducible nitric oxide synthase (iNOS), a M1 marker [24]. In agreement, our results showed that the subset of macrophages CD11b^+^ F4/80^+^ CD206^hi^ in C643 xenograft tumors also expressed high levels of Arg1 (Figure 3G) and low levels of iNOS (Figure 3H), whereas the population of macrophages defined as CD11b^+^ F4/80^+^ CD206^low/int^ expressed high levels of iNOS (Figure 3H) and low levels of Arg1 (Figure 3G). Similar results were observed in 8505C xenograft tumors. Since polarization of macrophages toward the M2 phenotype was associated with an increase in Arg1 level and a decrease in iNOS level in ATC xenograft tumors, we expressed the MFI ratio of Arg1/iNOS (M2/M1) as an indication of M2 macrophage polarization. Accordingly, in C643 xenograft tumors, the Arg1/iNOS ratio was 1.3 in the CD11b^+^ F4/80^+^ CD206^hi^ population, whereas this ratio was significantly lower in CD11b^+^ F4/80^+^ CD206^low/int^ macrophages (Figure 3I). Taken together, our results show the presence of macrophage infiltration with an M2-like skewed phenotype in ATC xenograft tumors, similar to that previously described in human thyroid tumors [13,14,15,16]. Thus, our data from the in vitro studies showing that ATC-cell-derived CM are capable of promoting M2-like macrophages in human monocyte THP-1 cells are consistent with our in vivo findings showing the presence of a large percentage of M2-like macrophage infiltration in ATC tumor xenografts.

### 3.2. TIM3-Positive Macrophages Were Recruited into ATC Xenograft Tumors

TIM3 was first identified as a surface molecule expressed on activated T-effector cells that acts as a negative regulator of the T-cell response [25]. Later, TIM3 was also found to be expressed on innate immune cells including monocytes and macrophages [25,26]. However, the precise role of TIM3 on macrophages in ATC remains unknown. Recently, we have reported high levels of TIM3 on THP-1 cells cultured with ATC cell-derived CM. Moreover, blockade of TIM3 reverses the macrophage polarization induced by ATC-cell-secreted soluble factors by using an in vitro model [18]. To further explore the in vivo levels of TIM3 on ATC tumors, we analyzed their expression on ATC-cell-induced tumors in mouse xenograft models. We first used RT-PCR analysis to detect TIM3 mRNA expression levels on ATC-cell-induced tumors. It has been reported that TIM3 is expressed on RAW264.7 cells [27]. In agreement, mRNA expression levels of TIM3 were detected in RAW264.7 cells (Figure 4A-I, lanes 1, upper panel). Similarly, we detected TIM3 mRNA expression levels in 8505C and C643 tumors and observed no significant differences between the two ATC-cell-induced tumors (Figure 4A-I, compare lanes 2–4 (8505C xenograft tumors) with lanes 5–7 (C643 xenograft tumors), upper panel). Figure 4A-II shows the quantitative data. To confirm these results, the expression levels of TIM3 in ATC xenograft tumors were also examined by immunofluorescence analysis of tumor sections. TIM3 is a protein with four defined regions: an immunoglobulin variable domain, a mucin domain, a transmembrane domain, and a cytoplasmic tail [25,26]. TIM3 expression was observed within 8505C xenograft tumors, indicating their presence within the TME. The protein is localized at cell surface and predominantly in the cytoplasm of cells (Figure 4B–D). Similar results were observed in C643 xenograft tumors (Appendix A). Then, and in order to validate our in vitro studies, we explored TIM3 protein expression on macrophages from ATC xenograft tumors. As shown in Figure 4E–G, flow cytometry analysis revealed that 8505C xenograft tumors exhibited 24% CD11b^+^ F4/80^+^ TIM3^+^ cells (Figure 4E), while 21% CD11b^+^ F4/80^+^ TIM3^+^ cells were found in C643 xenograft tumors (Figure 4F). No significant differences between tumors were observed when comparing TIM3 expression on macrophages (Figure 4G). Taken together, these results demonstrate, for the first time, that TIM3 is selectively expressed on macrophages in ATC in vivo.

### 3.3. CD45, CD68, and TIM3 Were Upregulated in Human ATC

We studied the clinical significance of our findings by analyzing public datasets. We compared the expression levels of different macrophage markers genes between 171 normal thyroid tissues and 52 ATCs. As shown in Figure 5A, and in line with our in vivo observations, ATCs exhibited a significantly higher expression of CD45 (leukocytes, PTPRC), CD68 (macrophages), and TIM3 compared with normal thyroids. Moreover, the expressions of CD45 (Figure 5B), CD68 (Figure 5C), CD206 (Figure 5D), CLEC7A (Figure 5E), and CD163 (Figure 5F) were highly correlated with TIM3 expression. Taken together, our results show that macrophage markers and TIM3 expression levels were increased in ATC tissues, thereby supporting our in vivo findings.

## 4. Discussion

ATC, although rare, is the most aggressive thyroid tumor with limited treatment options [3]. Recent genomic studies have shown that common genetic lesions are frequently found in TC, with BRAF and RAS mutations (including NRAS (≈19%), HRAS (≈5%), and KRAS (≈2%)) detected in ≈45% and ≈26% of ATC, respectively [28]. TAMs constitute more than 50% of infiltrating cells in ATC [13,14,15,16]. Consequently, targeting TAMs is considered a promising therapeutic approach for ATC. However, the way in which TAMs crosstalk with ATC cells remains elusive. In this study, we used two ATC-derived cell lines with distinct genetic lesions to establish tumors in immunocompromised NOD/SCID mice. 8505C cells harbor the *BRAF^V600E^* mutation, whereas C643 cells express the *HRAS^G13R^* mutation. Our choice of ATC cells to characterize ATC-infiltrating macrophages was based on the molecular profile of TC that has shown that BRAF and RAS mutations represent the two key oncogenic drivers in ATC [16,28]. Here, we observed differences in tumor growth along with significant differences in immune cell recruitment between both ATC cells (Figure 2A,B). These observations suggest the key role that the difference in the TME composition may play on tumor growth. However, the underlying mechanisms of this process are still poorly characterized. Unlike our findings, Morrison et al. found that C643 cells were not able to establish thyroid tumors in an orthotopic mouse model [29]. The discrepancies between our results and those observed in the study of Morrison et al. can be explained by different reasons. In addition to the different cell inoculation strategies (orthotopic vs. subcutaneous injection), the differences may be due to the murine models used in our studies and theirs. The athymic nude mice used for the establishment of orthotopic models of thyroid cancer differ in the degree of immune system compromise from the NOD/SCID mice used in our studies and therefore could explain, at least in part, the variation in the observed results. In spite of these distinct driver mutations, TIM3 expression on ATC-infiltrating macrophages was similar in both xenograft tumors, as shown in our in vivo analyses. These observations suggest that aberrant signaling initiated by different upstream driver mutations would not affect TIM3 regulation. However, whether BRAF and HRAS mutations play a role in TIM3 regulation on macrophages would be interesting to address in future studies.

Previously, we conducted functional in vitro studies that revealed that soluble factors produced and secreted by ATC cells elicit tumor-promoting M2-like macrophages, along with upregulation of the immune exhaustion marker TIM3 in THP-1 cells. Interestingly, using blocking antibodies to interrupt TIM3 signaling, a partial reversal of this effect was observed, suggesting the involvement of this protein on these effects [18]. Here, we validated our in vitro studies, demonstrating the presence of M2-like infiltrating macrophages in ATC-cell-induced xenograft tumors recapitulating the human disease. Significantly, we detected TIM3 expression on ATC-xenograft-tumor-infiltrating macrophages, identifying a new candidate immunotherapeutic target for the functional regulation of macrophages in the TME of ATC. Although we observed the expression of TIM3 in double-positive CD11b F4/80 cells, understanding the functional role of TIM3 in ATC-infiltrating macrophages would be important for designing novel strategies to target ATC. Whether TIM3 blockade alone or in combination with BRAF plus MEK inhibitors (currently FDA-approved for use in BRAF-mutated ATC) will eliminate tumor growth in vivo would be interesting to address in future studies.

Macrophages can regulate their phenotype by local tumor-derived signals. Multiple tumor-cell-derived signals including cytokines, chemokines, and growth factors, among others, recruit circulating monocytes as well as contribute to induce their M2-like phenotype polarization with pro-tumor functions in the TME. One of the best-validated cytokines involved in tumor progression is the pro-inflammatory cytokine interleukin 6 (IL-6). Interestingly, we have previously found that ATC cells secreted high amounts of IL-6 [7]. In addition, IL-6 plays a crucial role in macrophage activation. In this sense, the incubation of human monocytes with ATC-cell-derived CM promoted macrophage polarization, together with the increase in the expression and secretion of IL-6 in these cells. Therefore, IL-6 could represent a soluble factor in the media secreted by the ATC cells that, at least in part, contribute to the activation of macrophages. However, identifying the soluble factors released by ATC cells that propel the polarization of macrophages into an M2 phenotype would be interesting to address in future studies.

Until now, few studies have explored macrophage presence and polarization in ATC in vivo. Previously, the presence of M2-like macrophages in tumor xenografts derived from 8505C cells was evidenced [30]. In addition, a recent study revealed the presence of macrophage infiltration in xenografts derived from different thyroid cancer cell lines, including 8505C cells, by immunohistochemical assays [31]. In agreement with these observations, here, we demonstrated recruitment of macrophages on the basis of the expression of molecular marker including CD45, CD11b, and F4/80 using flow cytometry analysis and immunofluorescence assays. Furthermore, we demonstrated that these cells co-expressed high levels of CD206 and Arg1 with low levels of iNOS, indicating a preferentially M2-like phenotype. Future studies will be carried out in our laboratory to characterize the functionality of macrophage populations detected in our ATC xenografts.

Immunotherapy is being actively tested in patients with ATC [3]. In this regard, a recent clinical trial with anti-PD-1 antibodies showed an overall response rate of only 19% in ATC patients [19]. Therefore, there is a need to identify other co-inhibitory receptors to improve response rate of current immunotherapy for these patients. Recently, several novel immune checkpoints with therapeutic potential have been identified, with TIM3 being one of the most promising [25]. Solid evidence demonstrates that TIM3 inhibition enhances the anti-tumor effect of PD-1 blockade. Thus, a previous report showed that anti-TIM3 treatment of tumors in combination with anti-PD-L1 is more effective in controlling tumor growth when compared to treatment with anti-PD-L1 antibodies alone [32]. Additionally, the combination of TIM3 and PD-1 blockade enhanced cytokine production and proliferation of CD8 T cells isolated from patients with melanoma [33]. Currently, several clinical trials using the combination of anti-TIM3 and anti-PD-1/PD-L1 blocking antibodies are underway [25,34]. ATC is a type of thyroid cancer highly infiltrated with TAMs. Interestingly, data presented here provide evidence for the first time of the expression of TIM3 in TAMs in ATC xenograft tumors. Consequently, targeting immune checkpoints in TAMs, such as TIM3, should be pursued as a promising immunotherapeutic approach for ATC.

In this study, we used the immunocompromised NOD/SCID mice model, which is devoid of T and B lymphocytes. Thus, it would be interesting to address the interactions between tumor cells and macrophages in an immunocompetent host animal with an intact immune system that recapitulates the TME of the human disease. While these issues will be the subject of future research, this study is the first to identify the expression of the immune checkpoint inhibitor TIM3 on macrophages in ATC-cell-induced xenograft tumors.

## 5. Conclusions

In conclusion, our findings identified TIM3 as a new immunotherapeutic target in the TME of ATC xenograft tumors. Future studies will evaluate the potential clinical translational relevance of our findings to improve the current efficacy of immunotherapy in ATC patients.

## Figures and Tables

**Figure 1 biology-11-01609-f001:**
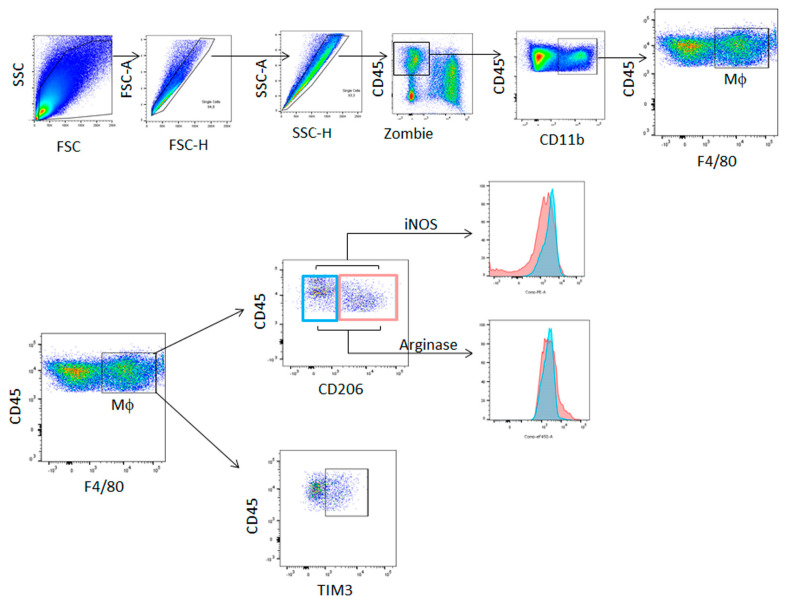
The gating strategy for flow cytometry analysis. A live cell gate was first applied on the basis of forward (FSC) and side (SSC) scatter, followed by doublets exclusion. Further analysis of living cells was achieved on the basis of aqua staining (Zombie dye). The expression of the different markers was performed in the Zombieneg CD45^+^ cells population.

**Figure 2 biology-11-01609-f002:**
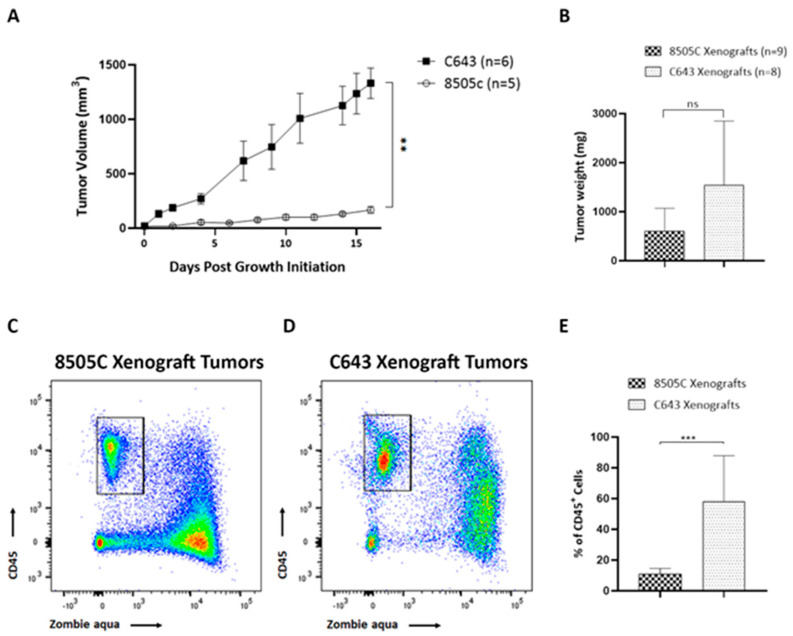
Recruitment of immune cells to *ATC xenograft tumors*. (**A**,**B**) Growth curves (**A**) and weight (**B**) of 8505C and C643 xenograft tumors. (**C**,**D**) Representative FACS analysis of live CD45^+^ (zombie aqua^neg^) cells (leukocytes) infiltrating 8505C xenograft tumors (**C**) and C643 xenograft tumors (**D**). (**E**) Quantification of the percentage of CD45^+^ cells detected out of all living cells (*n* = 8–10). Data are expressed as mean ± SEM of 3 independent experiments. ** *p* < 0.005, *** *p* < 0.0005.

**Figure 3 biology-11-01609-f003:**
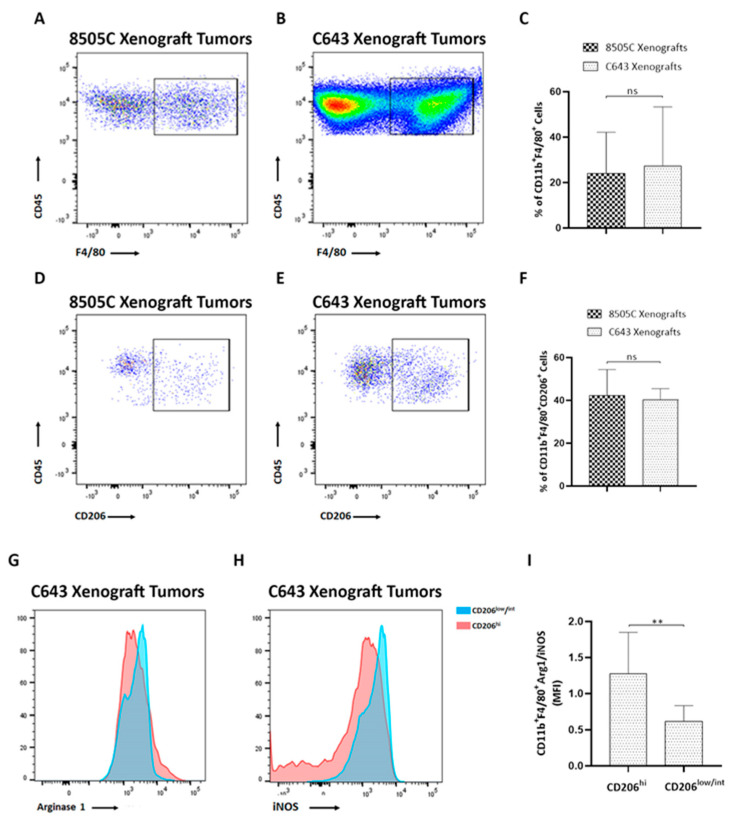
M2-like macrophages were recruited to ATC xenograft tumors. (**A**,**B**) Representative FACS analysis of macrophages (CD11b^+^ F4/80^+^) out of all CD45^+^ cells infiltrating 8505C xenograft tumors (**A**) and C643 xenograft tumors (**B**). (**C**) Quantification of the percentage of CD11b^+^ F4/80^+^ cells (*n* = 8–10). (**D**,**F**) Representative FACS analysis of M2-macrophages (CD11b^+^ F4/80^+^ CD206^+^) out of all CD45^+^ cells infiltrating 8505C xenograft tumors (**D**) and C643 xenograft tumors (**E**). (**F**) Quantification of the percentage of CD11b^+^ F4/80^+^ CD206^+^ cells (*n* = 3–5). (**G**,**H**) Representative histograms of Arg1 (**G**) and iNOS (**H**) expression in CD206^low/int^ (blue histograms) and CD206^high^ (pink histograms) populations in total cells from C643 xenograft tumors. (**I**) Mean fluorescence intensity (MFI) of Arg1/iNOS ratio in the CD206^high^ group and in the CD206^low/int^ group (*n* = 8–10). Data are expressed as mean ± SEM of 3 independent experiments. ** *p* < 0.005.

**Figure 4 biology-11-01609-f004:**
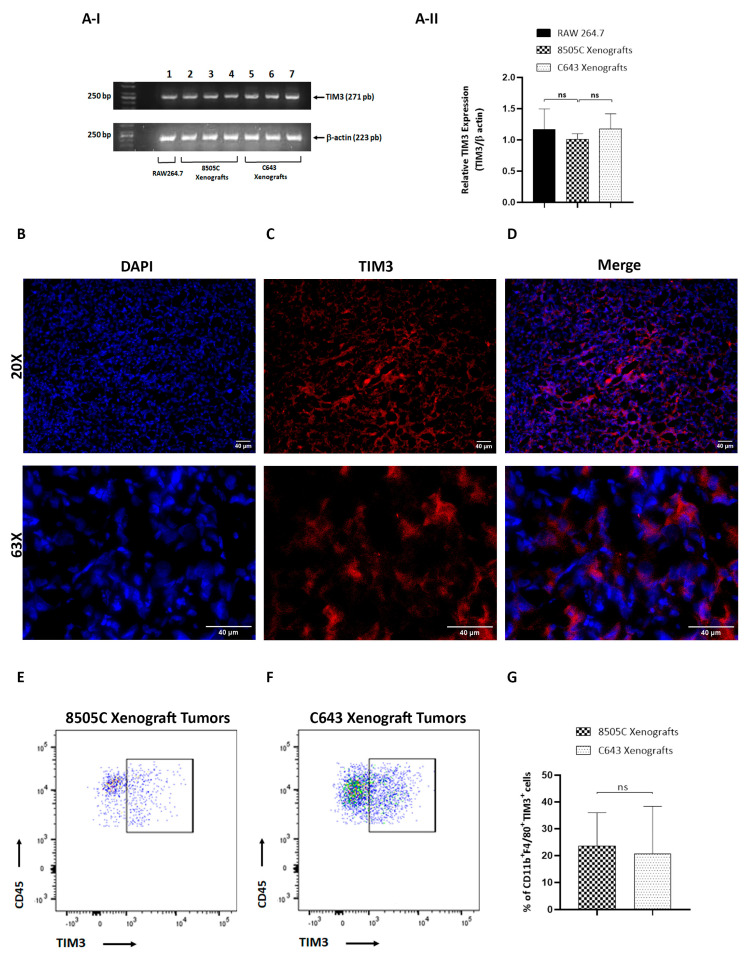
TIM3-positive macrophages were recruited to ATC xenograft tumors. (**A**) I. Upper panel: TIM3 mRNA expression in RAW264.7 cells and in ATC xenografts. Total RNA was isolated, and the mRNA expression of TIM3 was examined by RT-PCR analysis. Lane 1, RAW264.7 cells; lanes 2–4, 8505C xenograft tumors (*n* = 3); lanes 5–7, C643 xenograft tumors (*n* = 3). The lower panel represents the expression of actin. II. TIM3 mRNA expression was normalized by β-actin. Densitometry analysis of TIM3 is shown. Representative results of 3 independent experiments. (**B**–**D**) 20X and 63X images of 5 µm thick 8505C xenograft tumor sections immunostained with TIM3-PE antibody (red). Nuclei were counterstained with DAPI. Representative immunofluorescence images of 3 independent experiments. (**E**–**G**) Representative FACS analysis of 2 independent experiments of macrophages (CD11b^+^ F4/80^+^ TIM3^+^) out of all CD45^+^ cells in the 8505c xenograft tumors (**E**) and C643 xenograft tumors (**F**). (**G**) Quantification of the percentage of TIM3^+^ cells (*n* = 6–8).

**Figure 5 biology-11-01609-f005:**
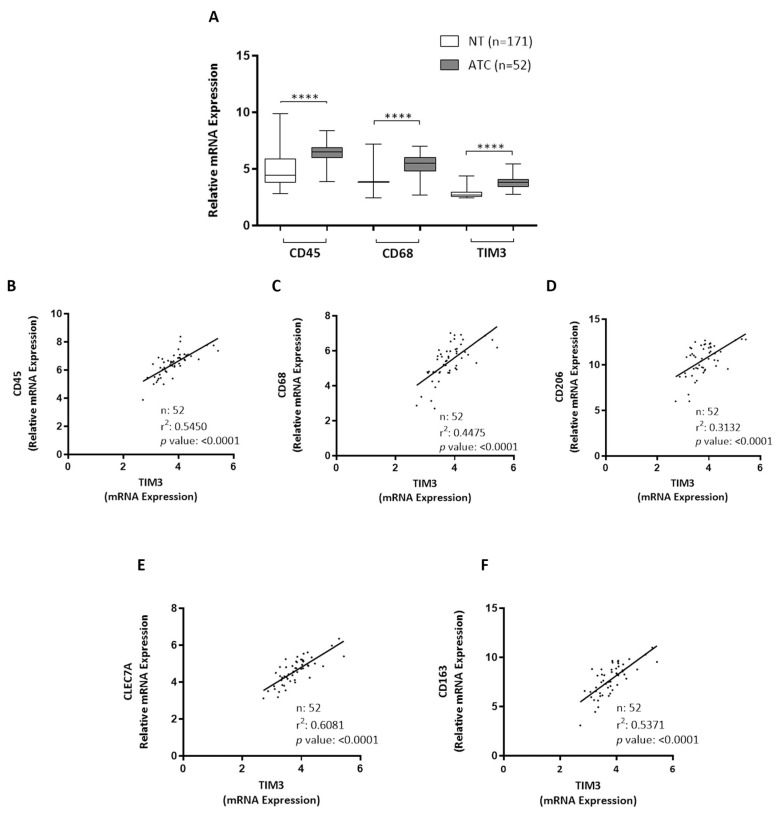
Macrophage markers, CD45 and CD68, and TIM3 expression in ATC. (**A**) Boxplots showing CD45, CD68, and TIM3 gene expression in normal thyroids (NT) and ATC samples derived from GEO datasets (GSE3467, GSE3678, GSE6004, GSE29265, GSE33630, GSE53157, GSE35570, GSE60542, GSE29265, GSE33630, GSE76039, and GSE65144). Statistical significance by Mann–Whitney test; **** *p* < 0.0001. (**B**–**F**) Correlation between expression of macrophages markers (CD45, CD68, CD206, CLEC7A, and CD163) and TIM3.

## Data Availability

The data presented in this study are available in the present manuscript and in the Appendix A.

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
