# Peer review of "TIM3 Expression in Anaplastic-Thyroid-Cancer-Infiltrating Macrophages: An Emerging Immunotherapeutic Target"

_biology, 2022, doi:10.3390/biology11111609_

Round 1

Reviewer 1 Report

Palacios and collaborators use a mouse xerograph model of anaplastic thyroid cancer. They aimed to demonstrate that TIM3 is a novel pharmacological target for treating this type of cancer.

The research design is not straightforward, some methods should be clarified, and more discussion is needed to improve their work. The following items should be addressed:

1.     The number of xenografts is too short. For example, in Figure A, C643 n=5 meanwhile 8505C n=3. How many mice are needed to achieve a global power of 80%? Are three mice enough to dissect differences?

2.     Consider the following:

The 8505C cell line is an adherent cell. Its doubling time is 36 hours. It carries BRAF(V600E) mutation.

The C643 cell line is an adherent cell too. But its doubling time is approximately 48 h hours. It carries an HRAS(G13R) mutation.

2.1 Please discuss the role of BRAF and HRAS in TIM3 expression.

2.2 Please discuss the differences between BRAF and HRAS in the thyroid cancer subtypes, which are their incidences in anaplastic thyroid cancer.

2.3 The mouse xenograft model is poorly described. How were mice anesthetized? Which is the specific anatomic area stated as the right flank?

2.4 In line 188, the authors stated, “Due to the difference in tumor growth between both groups, and to obtain comparable tumor sizes at the end of the experiments, 8505C cells were inoculated into mice two weeks before than C643 cells”. Authors need a titration cell curve to get the optimal density cell number. Please show the corresponding data.

2.5 Authors indicate no differences in tumor weight, despite that they indicate differences in tumor growth. Take into consideration that tumor growth is not only regulated by proliferation but also by matrix remodeling, stromal recruitment, immunoediting, and angiogenesis. Please refer to the Gompertz model, which indicates that growth rate decay is related to tumor enlargement. Moreover, an excess of cell densities could lead to improper angiogenesis, thus inducing cell death. Tumors need to be harvested before they necrotize. Did the authors do a time-lapse to determine the best time for harvesting? Please justify the time for harvesting and show the corresponding data

Mahdi Sohrabi-Haghighat & Atefeh Deris (2020) Growth rate, growth curve and growth prediction of tumour in the competitive model, Mathematical and Computer Modelling of Dynamical Systems, 26:2, 193-203, DOI: 10.1080/13873954.2020.1738498

Olague, Pedro Barragan and Kreinovich, Vladik, “Why Growth of Cancerous Tumors Is Gompertzian: A Symmetry-Based Explanation” (2016). Departmental Technical Reports (CS). 1073.

Gregório AC, Fonseca NA, Moura V, Lacerda M, Figueiredo P, Simões S, Dias S, Moreira JN. Inoculated Cell Density as a Determinant Factor of the Growth Dynamics and Metastatic Efficiency of a Breast Cancer Murine Model. PLoS One. 2016 Nov 7;11(11):e0165817. Doi: 10.1371/journal.pone.0165817. PMID: 27820870; PMCID: PMC5098815.

2.6  In Figures 2A and 2B authors showed differences among 8505C and C643 cell lines. But are quite the opposite of the results from Morrison JA and collaborators

Morrison JA, Pike LA, Lund G, Zhou Q, Kessler BE, Bauerle KT, Sams SB, Haugen BR, Schweppe RE. Characterization of thyroid cancer cell lines in murine orthotopic and intracardiac metastasis models. Horm Cancer. 2015 Jun;6(2-3):87-99. Doi: 10.1007/s12672-015-0219-0. Epub 2015 Mar 24. PMID: 25800363; PMCID: PMC4414894.

Please discuss the differences between the author’s experiment and Morrisons’ findings.

3.     In Figure 4A authors show an RT-PCR gel. Please state the replicates of this experiment, perform the densitometric assay, then graph and do the corresponding statistical analysis.

4.     In immunofluorescence images (Figure B-D) the figure bar is missing. Please add the corresponding images of the C643 cell line.

5.     In Figures 4E, F, and G authors show TIM3 expression. However, add at least one cell line of a different thyroid subtype to confirm that TIM3 is upper expressed in the anaplastic subtype.

6.     In line 364 authors stated, “soluble factors produced and secreted by ATC cells elicit tumor- promoting M2-like macrophages, along with the up-regulation of the immune exhaustion marker TIM3”. Please discuss in detail which soluble factors and how are regulated in the cell lines used. Which role plays BRAF and HRAS in TIM3 expression?

Reviewer 2 Report

The manuscript from Palacios et al provides an interesting study about the TIM3 expression in Anaplastic Thyroid Cancer (ATC) associated macrophages. Most of the claims in the manuscript are based on experimental observations. However, before publication, the manuscript should be benefited by addressing the following comments.

Major comments:

1. More details are required why the authors used 805C and C643 ATC cells. Apart from different sources, was there anything that explains their different tumor growth curves?

2. In the summary, abstract, or introduction: details are required to make the manuscript easy to understand for people outside the field. For instance, what are THP-1 cells in summary (line-23), and what is the significance of Arg1/iNOS ratio in the abstract (line 41)?

3. Since the major claim of the manuscript is about TIM3 expression, high magnification images are required in Figure 4B-D. Control image is also required for cells without inoculation of 8505C. Currently, 20 X images are shown, it would be appropriate to show at least 40x images with a better explanation of the cellular localization of TIM3.

Reviewer 3 Report

The research article by Luz et al., describes the plausibility of using TIM3 as an emerging immunotherapeutic target in Anaplastic thyroid cancer. In this article, authors have used two types of ATC cell lines to induce xenograft tumors in a murine model. They depicted TIM3 expression in the xenografts and increased number of TAMs in the tumors. However, the manuscript needs to undergo revision before getting published.

Here are my comments:

1.       All the procedures and bioinformatic analysis performed by the authors were well described. However, the results are very descriptive. Inferences/observations need to be emphasized.

2.       Results section 3.2 and Fig 4 is ambiguous. TIM3 expression is present, but the experiment does not reveal the expression was on macrophages. The control in QPCR analysis i.e R264.7 does not suffice the findings. It only reveals macrophages express TIM3, but not on TAMS. I suggest the authors to perform a co-localization experiment of macrophages marker and TIM3 in immunofluorescence.

3.       If authors pave the argument of TIM3 as an immunotherapeutic agent, there should be a negative/silencing control to the xenograft model. An In vivo silencing of TIM3 will confirm the role of TIM3 as an immunotherapeutic agent of ATC.

 Manuscript needs to be checked for repetitions.

Reviewer 4 Report

The manuscript is lucid and well-written.

In this manuscript, authors have extended their previous findings published in Stempin CC., et al.,  to validate into in vivo system by targeting Anaplastic Thyroid Cancer cell (8505C and C643 cells) induced xenograft tumor models using RT-PCR, flow cytometry, immunofluorescence staining, and gene expression analysis. Their major findings include:

A large scale of M2-like macrophages infiltration and the expression of TIM3 (T cell immunoglobin and mucin-domain containing protein-3) in ATC tumor xenografts. Since TIM3 is an immune checkpoint in macrophages thus their finding sets a great platform for further research to provide a promising immunotherapeutic target for ATC patients.

Minor comments:

1.     Most of the curve axes have very poor font size, particularly Figure-1, enlargement of these will make better visualization.

2.     Line-153, need to put space, “50 mM”

3.     Figure-4 and line-291-292; Immunofluorescent staining is shown for TIM3 and F4-80-PE with only 8505C xenograft tumor. My suggestion is to show immunostaining images for the 643 xenograft tumor model as well in supplementary.

Round 2

Reviewer 1 Report

Palacios and collaborators used a mouse xerograph model of anaplastic thyroid cancer. Demonstrating that TIM3 is expressed by infiltrating macrophages. They also suggest that it could be a novel pharmacological target for thyroid cancer.

The authors discussed some concerning aspects of methodology and enriched their discussion.

Reviewer 2 Report

All my comments are addressed!!!

The manuscript is now improved for publication.

Reviewer 3 Report

Authors have modified the manuscript according to the suggestions. 

Reviewer 4 Report

I thank the authors for addressing my previous comments and all the amendments.

The authors have sufficiently improved their manuscript.